# Respiratory Syncytial Virus Matrix Protein Is Sufficient and Necessary to Remodel Host Mitochondria in Infection

**DOI:** 10.3390/cells12091311

**Published:** 2023-05-04

**Authors:** MengJie Hu, Marie A. Bogoyevitch, David A. Jans

**Affiliations:** 1Department of Biochemistry and Molecular Biology, Monash University, Melbourne, VIC 3800, Australia; mengjie.hu000@gmail.com; 2Department of Biochemistry and Molecular Biology, University of Melbourne, Melbourne, VIC 3010, Australia

**Keywords:** respiratory syncytial virus (RSV), RSV matrix protein (M), host cell mitochondria, RSV infection

## Abstract

Although respiratory syncytial virus (RSV) is the most common cause of respiratory infection in infants, immunosuppressed adults and the elderly worldwide, there is no licensed RSV vaccine or widely applicable antiviral therapeutics We previously reported a staged redistribution of mitochondria with compromised respiratory activities and increased reactive oxygen species (ROS) generation during RSV infection. Here, we show for the first time that the RSV matrix protein (M) is sufficient and necessary to induce these effects. Ectopically expressed M, but not other RSV proteins, was able to induce mitochondrial perinuclear clustering, inhibition of mitochondrial respiration, loss of mitochondrial membrane potential (Δψ_m_), and enhanced generation of mitochondrial ROS (mtROS) in infection. Truncation and mutagenic analysis revealed that the central nucleic acid-binding domain of M is essential for the effects on host mitochondria, with arginine/lysine residues 170/172 being critically important. Recombinant RSV carrying the arginine/lysine mutations in M was unable to elicit effects on host mitochondria. Further, wild-type but not mutant RSV was found to inhibit the mRNA expression of genes encoding mitochondrial proteins, including Complex I subunits. Importantly, the RSV mutant was impaired in virus production, underlining the importance of M-dependent effects on mitochondria to RSV infection. Together, our results highlight M’s unique ability to remodel host cell mitochondria and its critical role in RSV infection, representing a novel, potential target for future anti-RSV strategies.

## 1. Introduction

Respiratory syncytial virus (RSV), an enveloped RNA virus of the *Pneumoviridae* family in the order of *Mononegavirales*, is the single most common cause of acute lower respiratory tract infection in young children, elderly, and immunosuppressed individuals, with a global incidence of 64 million infections and c. 250,000 deaths/year [1,2]. Despite this prodigious burden of RSV infection, there is no efficacious, widely applicable antiviral strategy to combat the virus [3,4] (see Section 4), making detailed understanding of the RSV–host interface of paramount importance. Like all negative-strand pneumoviruses, RSV’s polymerase complex is packaged in the virion, allowing viral transcription or replication to proceed in the cytoplasm of the infected cells, followed by virion assembly and budding [5]. The RSV matrix protein (M) is present within the virion, playing key roles in the virus life cycle in the host cell [6,7,8]; early in infection, M is transported into the nucleus by the host nuclear transport protein importin β1 [8,9], serving a dual role of inhibiting host cell transcription, as well as avoiding the potential suppression of viral transcription in the cytoplasm by M [8,10]. We recently showed that the effects on host transcription, including of nuclear genes encoding mitochondrial components, are strongly dependent on the association of the M protein with host chromatin; mutations in RSV M reducing chromatin association severely impact RSV infectious virus production [8]. Later in infection, M traffics to the cytoplasm through the action of the nuclear export protein Crm1, associated with inclusion bodies, sites of RSV transcription and replication, before coordinating viral assembly and budding at the plasma membrane [7,8,11,12,13]. To promote virion assembly and budding, M has been shown to recruit cellular factors, including the mitochondrial proteins implicated in mitochondria-mediated stress response and apoptosis [14,15]. Our more recent studies on RSV-infected cells have revealed a staged redistribution of mitochondria, with compromised respiratory activities and increased reactive oxygen species (ROS) generation [7,16,17], correlating with the transcriptional inhibition of host genes [8]. However, the dependence on particular viral components of RSV’s impact on host mitochondria has not been determined.

The present study examines the interaction between RSV M and host cell mitochondria for the first time. Using a combination of redox/membrane potential sensitive dyes, high-resolution imaging/flow cytometric analysis, reverse transcription quantitative PCR, and bioenergetics analyses, proved that it was possible to establish that ectopically expressed M, but not other RSV proteins, is able to induce mitochondrial perinuclear clustering, inhibition of mitochondrial respiration, loss of Δψ_m_, and enhanced generation of mitochondrial ROS (mtROS) in infection. Truncation and mutagenic analysis revealed the central nucleic acid-binding domain of M to be key to the effects on host mitochondria, with arginine/lysine residues 170/172 being critically important. Recombinant RSV carrying the arginine/lysine mutations in M was unable to elicit effects on host mitochondria. Strikingly, infection with wild-type but not mutant RSV resulted in reduced mRNA levels for genes encoding mitochondrial proteins, including Complex I subunits [8,16,17]. Importantly, the RSV mutant virus is also impaired in virus production, underlining the importance of M-dependent effects on mitochondria to RSV infection. Together, the results highlight M’s unique ability to remodel host cell mitochondria and its key contribution to RSV infection, representing a novel target for the development of anti-RSV strategies.

## 2. Materials and Methods

### 2.1. Cell Culture, RSV Infection and RSV Growth

Cell lines were cultured in a humidified atmosphere (5% CO_2_, 37 °C) and passaged at 3-day intervals by dissociation using trypsin/EDTA (Gibco, New York, NY, USA). A549 cells (human adenocarcinoma alveolar basal epithelial cells) were maintained in Ham’s F-12K (HF-12K) medium containing 10% heat-inactivated foetal calf serum (FCS; DKSH Australia Pty Ltd., Mulgrave, Australia) and 100 U/mL penicillin and streptomycin (Gibco), 2 mM L-glutamine (Gibco) and 1.5 g/L sodium bicarbonate. Vero cells (African green monkey kidney epithelial cells) were cultured in Dulbecco’s modified Eagle’s medium (DMEM, Gibco) containing 10% FCS.

Stocks of recombinant wild-type and mutant RSV virus were grown in Vero cells as previously [8,18]; all stocks were titrated by plaque assay to determine the plaque forming units (PFU) so that an identical multiplicity of infection (MOI) of 1–3 could be used in all experiments. A549 cells were grown for 12 h prior to infection in HF-12K medium containing 2% FCS. After 2 h, cells were washed and media replaced; cell-associated viruses at various times post infection (p.i.) were retained for analysis of viral genomes (by quantitative PCR) or infectious virus (PFU) as per [18].

### 2.2. Plasmid Construction and Transfection

Plasmids encoding RSV -M, -F, -P, or -N for mammalian expression have been described [19]. Plasmids encoding truncations of M (1–110, 1–183, 183–256, and 110–183) fused to GFP have been previously described [8,9]. Plasmid encoding point mutants of M (K121A, K130A, K156T/K157T, R170A, R172A, R170A/K172A, R170T/K172T) fused to GFP were prepared by Dr. Hong-mei Li (Monash University, Melbourne, Australia) using the Gateway^TM^ system (Invitrogen, Waltham, MA, USA).

Cultures of cells grown overnight on glass coverslips were transfected to express various RSV proteins or M mutants, using Lipofectamine 3000 (Invitrogen). Cells were cultured for 24 h prior to immunofluorescence analysis.

### 2.3. Immunofluorescence and Confocal Scanning Laser Microscopy (CLSM)

Mock-, rA2- or rA2:Mmut-infected A549 cells were stained with MitoTracker Deep Red or MitoTrackerRed CMXRos (ThermoFisher Scientific, Waltham, MA, USA; 100 nM, 15 min) prior to fixing, washing, and staining, as described previously [20]. The primary antibodies used were anti-RSV antibody (1:400, ab20745, Abcam, Cambridge, UK), anti-FLAG antibody (1:100, F1804, Sigma, St. Louis, MO, USA) with dye-tagged secondary antibodies (anti-goat Alexa Fluor 488, 1:1000, A11055, or anti-mouse Alexa Fluor 488, 1:1000, A32723, ThermoFisher Scientific) as appropriate. In all analyses of fixed cells, the nuclei were stained with DAPI (1:15,000 in PBS, 10236276001, Sigma). After mounting onto glass slides with Biomedia Gel Mount (ProSciTech, Kirwan, Australia), cells were imaged using a Leica TCS SP5 channel confocal and multiphoton microscope (63X objective, oil immersion). Images (512 × 512 pixels, 8- or 12-bit) were collected and viewed using the Leica Application Suite Advanced Fluorescence Lite Version: 2.8.0 build 7266 viewer software.

### 2.4. Quantitative Analysis of Mitochondrial Organisation/Distribution

Quantitative analysis of mitochondrial organisation and distribution was performed using custom scripts programmed in Python using *numpy* [21], *scipy*, *scikit-image* [22], *matplotlib* [23], and *seaborn* [24,25] (see https://gitlab.erc.monash.edu.au/mmi/mito (accessed on 28 February 2023), as previously described [16]). To quantify mitochondrial perinuclear distribution, the radius of the circle required to enclose 90% of the MitoTrackerRed fluorescence relative to the centre of the nucleus was determined (R_90%_) [26]; nuclei were segmented using a 2-pixel Gaussian filter and an Otsu threshold [27] to the DAPI channel (objects < 500 pixels excluded). Mitochondria were segmented by applying a 2-pixel Gaussian filter and a Li threshold [28] to the MitoTracker Red channel (objects < 10 pixels excluded). Adjacent cells were split using a Watershed transform, where nuclei centroids situated ≥ 10 pixels apart were used as markers. The infected cells analysed were those with mean anti-RSV-Alexa Fluor 488 fluorescence intensity 5–10 arbitrary units (A.U.). The R_90%_ was calculated by creating a Euclidean distance map using the nuclei centroids, which was masked using the segmented mitochondrial region to generate a map where the intensity of each pixel represents the distance of that pixel from the centre of the nucleus (i.e., the radius). A cumulative histogram was constructed from pixel radial distances, and R_90%_ was then calculated as the radius accounting for 90% of the mitochondrial pixels.

### 2.5. Assessment of Mitochondrial Bioenergetics and Function

The OCR (oxygen consumption rate) and ECAR (extracellular acidification rate) were quantified using the Seahorse XF96 Extracellular Flux Analyser (Seahorse Biosciences, North Billerica, MA, USA) [20]. A549 cells were plated (1.5 × 10^4^ cells/well, 10% FCS/HF-12K) with or without virus (rA2 or rA2:Mmut) infection (MOI 3, 2% FCS/HF12K). Cells were washed twice with pre-warmed XF assay buffer (unbuffered DMEM containing 25 mM glucose, 2 mM L-glutamine, and 1 mM sodium pyruvate, pH 7.4) and then equilibrated in XF buffer (37 °C, 1 h) before Seahorse measurement at 24 h. Respiratory parameters for basal, ATP-linked, maximal uncoupled, spare, and non-mitochondrial respiration were calculated from OCR in response to the sequential additions of 1 μM oligomycin (ATP synthase inhibitor), 1 μM FCCP (carbonyl cyanide p-trifluoromethoxyphenylhydrazone, proton ionophore), and a combination of 1 μM antimycin A (Complex III inhibitor), and 1 μM rotenone (Complex I inhibitor), respectively [20].

### 2.6. Measurement of Mitochondrial Membrane Potential (Δψ_m_) and mtROS

Δψ_m_ was determined using Δψ_m_-sensitive fluorescent dyes. In the case of tetramethylrhodamine ethyl ester (TMRE) [29], A549 cells were mock- or eGFP-RSV-infected (MOI 3) for 12–24 h with TMRE (ab113852, abcam; 50 nM; Ex/Em: 561/565 ± 25nm) and Hoechst (H3570, ThermoFisher Scientific; 5 µg/mL; Ex/Em: 405/470 nm) added for the last 15 and 5 min, respectively, in the dark before live imaging as per [16]. To minimise photobleaching and phototoxicity to cells during imaging, live cell imaging was performed using a CLSM with 8 kHz resonant optical scanners (resonant scanning CLSM) for image resolution (512 × 512 pixels, 12-bit). To quantify the fluorescence levels of TMRE in cells, single in-focus planes were analysed for area, integrated density, and mean gray value using Fiji (https://fiji.sc/ accessed on 28 February 2023). The corrected total cell fluorescence (CTCF) was calculated using the formula CTCF = integrated density—(area of selected cell × mean fluorescence of background readings) [30,31]. The averages of the total corrected fluorescent intensity values of 15–30 cells were calculated for each treatment condition. For FACS analysis, transfected cells were trypsinised at different times, centrifuged, resuspended in FACS buffer (2% heat-inactivated FCS, 10 mM HEPES [(4-(2-hydroxyethyl)-1-piperazineethanesulfonic acid], 2 mM L-Glutamine, 2 mM EDTA solution) containing TMRE (50 nM, 37 °C, 15 min), and then analysed using a BD LSRII flow cytometer. Data analysis was performed using FlowJo software (Tree Star, Inc, Ashland, OR, USA).

mtROS were detected using the mitochondrial targeting ROS sensor, flavin-rhodamine redox sensor 2 (FRR2) as per [16]. Mock-infected A549 cells were treated with rotenone (0.5 μM, 30 min), mitoquinone mesylate (MitoQ, provided by Health Manufacturing, New Zealand); 1 μM, 2 h) [32], rotenone (0.5 μM, 2 h) followed by MitoQ (1 μM, 2 h), or DMSO (vehicle) as a control; rA2- or rA2:Mmut-infected (MOI 3) cells were treated with or without MitoQ (1 μM) for the last 2 h. All cells were incubated with MitoTracker Deep Red (M22426, ThermoFisher Scientific; 100 nM; Ex/Em: 633/665 nm) and FRR2 (2 μM) for 15 min, with Hoechst (5 µg/mL) added at the last 5 min in the dark before imaging using resonant scanning CLSM at 12 h. The ratiometric output of FRR2 [33] (I _(Ex514)_/I _(Ex488)_; the ratio of the intensity of red emission [denoted as I] at 580 ± 20 nm upon excitation [Ex] at 514 nm compared to 488 nm) is an indicator for mtROS accumulation. Ratiometric I _(Ex514)_/I _(Ex488)_ images were generated by a pixel-wise division of the 514 nm and 488 nm emission image channels using Fiji (https://fiji.sc/ accessed on 28 February 2023). For all samples, images were set to 32-bit float precision with a display range of min = 0.0 and max = 15.0 to facilitate comparison). To quantify the mitochondrial-localised ratio, a CellProfiler pipeline (http://cellprofiler.org/ accessed on 28 February 2023) was set up, whereby a pixel-wise image of I _(Ex514)_/I _(Ex488)_ was derived by a pixel-wise division of the emission image channels acquired at 514 nm and 488 nm excitation and stored as a 32-bit float image. Regions containing mitochondria were then segmented from the MitoTracker Deep Red channel by applying a 5-pixel Gaussian blur and an Otsu auto-threshold [27], and then filtered to exclude all regions that were < 1000 pixels. These regions were then used to determine the mean ratiometric pixel value using the I _(Ex514)_/I _(Ex488)_ image above.

Changes in cellular ROS production were detected using dichlorodihydrofluorescein diacetate (H_2_DCFDA/DCF, D399, Thermo Scientific) [34]. A549 cells were mock-, rA2- or rA2:Mmut-infected (MOI 3), or treated with rotenone (0.5 μM, 30 min), followed by MitoQ (1 μM, 2 h) or DMSO, or treated with DMSO (vehicle) as a control. Cells were then incubated with MitoTracker Deep Red (100 nM, 15 min), with Hoechst and DCF (2.5 µM; Ex/Em: 496/517–527 nm) added at the last 5 min in the dark before imaging using resonant scanning CLSM at 12 h p.i. as per [16], with fluorescence levels of DCF in cells analysed as per the CTCF quantification for TMRE.

### 2.7. Quantitative PCR (qPCR) to Quantify Viral RNA

The number of viral RNA genomes was estimated using qPCR. RNA was extracted from the cell-associated virus using the Isolate II RNA kit (Bioline), as per the manufacturer’s instructions. Nucleotide sequences (primers and probe) were directed to a region within the RSV nucleoprotein (N) gene [35]. Primers (forward 5′ CTC AAT TTC CTC ACT TCT CCA GTG T 3′; reverse 5′ CTT GAT TCC TCG GTG TAC CTC TGT 3′) were synthesised by Integrated DNA Technologies. The probe (5′ TCC CAT TAT GCC TAG GCC AGC AGC A 3′) was labelled with the 5′ reporter dye 6-carboxy-fluorescein (FAM) and the 3′ quencher dye 6-carboxy-tetramethylrhodamine (TAMRA, Applied Biosystems, Waltham, MA, USA). A one-step protocol was used with 10 μL of RNA added to each reaction mixture containing 1X TaqMan Fast Virus 1-Step Master Mix (Applied Biosystems), 300 nM of each of the primers, and 200 nM of the probe. The amplification profile used was 1 cycle for 5 min at 50 °C and 20 sec at 95 °C, followed by 40 cycles for 3 and 30 sec at 95 °C and 60 °C, respectively. Genome copy numbers were determined by extrapolation from a standard curve produced using a plasmid carrying the RSV N gene [36].

### 2.8. Reverse Transcription qPCR (RT-qPCR) to Quantify Host Gene mRNA Levels

Host gene mRNA levels were analysed in infected cells by RT-qPCR. Total RNA was extracted from A549 cells infected with rA2 or rA2:Mmut using the Isolate II RNA kit (Bioline). cDNA was synthesised from 500 ng of total RNA using Superscript III First-Strand Synthesis System for RT-PCR (Invitrogen) with Oligo(dt)20 primers. RT-qPCRs for host genes were performed using SYBR green (Sensimix SYBR Hi-ROX, Cat. No. QT605-05, Bioline). For gene primers, RPLPO (Large ribosomal protein): forward 5′ CTG GAA GTC CAA CTA CTT CCT 3′; reverse 5′ CAT CAT GGT GTT CTT GCC CAT 3′. HPRT1 (Hypoxanthine phosphoribosyltransferase 1): forward 5′ TGT CAT GAA GGA GAT GGG AG 3′; reverse 5′ CAG TCT GAT AAA ATC TAC AGT C 3′. LIPT2 (Lipoyl(Octanoyl) Transferase 2): forward 5′ CAC GAT GTG CTC AAA CCA C 3′; reverse 5′ CTA GAC GAT CGC AAG ATC TGC 3′. CYP24A1 (Cytochrome P450 Family 24 Subfamily A Member 1): forward 5′ GTG ACC ATC ATC CTC CCA AA 3′; reverse 5′ AGT ATC TGC CTC GTG TTG TAT G 3′. PDK4 (Pyruvate Dehydrogenase Kinase 4): forward 5′ CAT CTG GGC TTT TCT CAT GGA 3′; reverse 5′ TCC CGA CCC AAT TAG TAA ATA CC 3′. NDUFA10 (NADH: Ubiquinone Oxidoreductase Subunit A10): forward 5′ CAC AAA AAG CAA TCC ATC CTG 3′; reverse 5′ AGG GTT CCA AAC ATC CAG AA 3′. NDUFV1 (NADH: Ubiquinone Oxidoreductase Core Subunit V1): forward 5′ CGG GTA TCT GTG CGT TTC AG 3′; reverse 5′ GGT CTT CAT CCT TCA GCG AG 3′. NDUFC2 (NADH: Ubiquinone Oxidoreductase Subunit C2): forward 5′ ATT GAT AAC CTA ATC CGG CGG 3′; reverse 5′ TCC AAA CAT TTC ACG GTC CC 3′. NDUFB10 (NADH: Ubiquinone Oxidoreductase Subunit B10): forward 5′ CCC AAT CCC ATC GTC TAC ATG 3′; reverse 5′ GTA ATA CCT GTT CTT TGC GTG C 3′. ACAD9 (Acyl-CoA Dehydrogenase Family Member 9): forward 5′ GTC GGA GAT GGG TTT AAG GTG 3′; reverse 5′ TCA CTG AGC CTC TTG TTA AAC TG 3′. PRDX5 (Peroxiredoxin-5): forward 5′ GCA GTG AAG GAG AGT GGG 3′; reverse 5′ GGA CAC CAG CGA ATC ATC TAG 3′. DYNC1H1 (Dynein cytoplasmic 1 heavy chain 1): forward 5′ GCG TAT TTC AAG TCC TGT ACC T 3′; reverse 5′ GAC AAT AAG CTC CTA ACT TTG CC 3′. FOXO3 (Forkhead box protein O3): forward 5′ CTC GGC GAA GGA GAA GC 3′; reverse 5′ GAG GAG GAA TGT GGA AGG TG 3′. CENPU (Centromere protein U): forward 5′ TCC AGC CTT CCA GCT CTG TT 3′; reverse 5′ AGC TTC TCT AAC TGA TGG TTG A 3′. The RT-qPCR reaction conditions were as follows: 95 °C for 10 min, followed by 40 cycles of 30 s each of 95, 60, and 72 °C. The termination involved one cycle of 15 s each of 95, 60, and 95 °C. The copies of the host genes from the virus-infected cells were calculated relative to those of mock-infected cells.

### 2.9. Statistical Analysis

All quantitative data presented in this study correspond to the mean value of *n* experiments ± SEM for *n* ≥ 3. Significance levels were determined by performing a two-tailed *t* test or ANOVA (GraphPad Prism 6) where applicable, with a value of *p* ≤ 0.05, considered to be statistically significantly different.

## 3. Results

### 3.1. RSV Matrix Protein (M) Is Sufficient and Necessary to Induce Mitochondrial Perinuclear Clustering

We previously showed that RSV infection results in the subcellular redistribution of mitochondria, concomitant with effects on mitochondrial function [16,17]. As a first step to determine whether particular RSV proteins may contribute, out of the context of viral infection, to the effects on mitochondria in RSV infection, RSV M (matrix), F (fusion), P (phosphoprotein), and N (nucleoprotein) were ectopically expressed in A549 cells, and imaged by high-resolution CLSM (Figure 1A). All transfected cells exhibited a mixture of tubular and punctate mitochondrial structures stained by MitoTrackerRed (Figure 1A). In contrast to the even distribution of mitochondria throughout cells expressing F, P, or N (Figure 1A, last three rows), substantial perinuclear mitochondrial clustering was observed in cells transfected to express M (Figure 1A, second row). Quantitative analysis of the R_90%_ parameter, the radius of a circle required to enclose 90% of the MitoTrackerRed fluorescence relative to the centre of the nucleus [26], revealed significantly (*p* < 0.001) reduced R_90%_ (over 40%) in M-expressing cells compared to cells expressing any of the other viral proteins, or mock-transfected cells (Figure 1B), implying that RSV M may be sufficient to induce the perinuclear clustering of mitochondria observed in RSV-infected cells.

As a first step to confirm these results, we employed FLAG-tagged RSV M constructs, one of which, FLAG-RSV Mmut, retains dual point mutations (R170T/K172T) previously shown to abrogate nuclear chromatin association/transcriptional inhibition by M, and in the context of the recombinant RSV virus, result in significant attenuation [8]. Cells were transfected to express FLAG-tagged RSV M with or without the mutation; the transient expression of RSV M mut showed an equal distribution of elongated mitochondria throughout the cells (Figure 1C, bottom row) in contrast to the perinuclear accumulation of mitochondrial puncta in RSV Mwt-expressing cells (Figure 1C, second row). Quantitative analysis for the R90% parameter supported this observation (Figure 1D), confirming that ectopically expressed M requires residues R170/K172 for perinuclear mitochondrial clustering, reinforcing the idea that wild-type M is sufficient to cause the perinuclear clustering of mitochondria seen in RSV infection [16,17]. Importantly, it represents the first indication that M’s effects on host transcription [8] may be key to these effects.

To confirm which domains of M are responsible for mitochondrial perinuclear clustering, a series of previously characterised truncation derivatives of M fused to GFP [8] (Figure 2A left) were ectopically expressed as previously in A549 cells (Figure 2A right). While the GFP-RSV-M-1-183 and -110-183 truncation derivatives (fourth and sixth rows) were observed to induce perinuclear accumulation of mitochondrial puncta to the same extent as the wild-type RSV M (WT, second row), both the GFP-RSV-M-1-110 and -183-256 (third and fifth rows) resulted in an equal distribution of elongated mitochondria comparable to that of the mock-transfected cells (first row). Quantitative analysis (Figure 2B) confirmed the findings, revealing significantly (*p* < 0.001) reduced (over 40%) R_90%_ in cells expressing wild-type M, M-1-183, and -110-183 compared to cells expressing any of the other M truncation derivatives, or mock-transfected cells, supporting the conclusion that the central arginine/lysine-rich nucleic acid-binding domain of M (residues 110–183) is essential for mitochondrial perinuclear clustering.

To determine the key residues in the central domain responsible for mitochondrial distribution in cells, we expressed M with point mutation(s) in central domain arginine or lysine residues (Figure 2C, left) [8]. The ectopic expression of the K121A, K130A, or K156T/K157T substituted derivatives of GFP-M, which all resulted in mitochondrial perinuclear clustering, essentially identical to that of WT GFP-M (Figure 2C, right, second–fourth rows compared to first row), as supported by the quantitative analysis of the R_90%_ parameter (Figure 2D). Intriguingly, the substitution of arginine at position 170 or 172 resulted in reduced mitochondrial perinuclear clustering (Figure 2C, right, fifth and sixth rows). Strikingly, the double mutant derivatives, R170A/K172A or R170T/K172T (Mmut), completely lacked effects on mitochondrial distribution; their mitochondrial morphology resembling that of mock-transfected cells (Figure 2C, the last two rows). Analysis for the R_90%_ parameter supported this observation (Figure 2D), confirming that mutation of R170/K172 eliminates the ability of M to induce perinuclear mitochondrial clustering in transfected cells, consistent with the idea that M is sufficient for the effects on mitochondrial organisation.

To test the relevance of the above results to RSV infection, a previously characterized recombinant mutant virus (rA2:Mmut), containing the specific R170T/K172T substitutions within M [8], was compared to WT virus. We examined mitochondrial morphology in A549 and Vero cells infected with the wild-type and mutant viruses at different p.i. times. In stark contrast to the striking perinuclear clustering of mitochondria during rA2 infection in both lines (Figure 3A and Appendix A, second and third rows of panels), rA2:Mmut-infected cells showed dispersed mitochondrial distribution completely comparable to uninfected cells for both lines (Figure 3A and Appendix A, compare the last two rows with the first row of panels). The results for the R_90%_ parameter (Figure 3B and Appendix A) supported these observations, showing restored mitochondrial distribution following rA2:Mmut infection. Clearly, the R170T/K172T substitutions within the M abrogate mitochondrial perinuclear clustering in the context of virus infection is consistent with the idea that M is the key component of RSV responsible for the effects on mitochondrial distribution. To confirm that the inability to remodel the host mitochondria impacted the RSV infectious cycle, RSV virus production by rA2:Mmut was compared to that for WT rA2. As previously described in [8], the mutant virus showed significantly (*p* < 0.001) reduced (over 60%) virus production in terms of RNA genomes and infectious virus in both A549 (Figure 3C,D) and Vero (Appendix A) cells at 24 h p.i. compared to wild-type rA2 virus. The fact that the rA2:Mmut virus shows reduced viral replication and production underlines the importance of M-dependent mitochondrial redistribution to RSV infection. Strikingly, the rA2:Mmut virus has also been shown to be impaired in transcriptional inhibition, including of genes encoding mitochondrial components (see ref. [8]), implying a potentially strong link between effects at the level of the nucleus and those at the level of mitochondria (see below).

### 3.2. RSV M Is Key to Inhibition of Mitochondrial Respiration in Infection

We previously showed that RSV infection impairs mitochondrial respiration [16,17]. To examine the M dependence of this effect, cells infected with rA2 or rA2:Mmut were compared for effects on mitochondrial respiration at 24 h p.i. using the Seahorse XF96 Extracellular Flux Analyser. The oxygen consumption rate (OCR) and extracellular acidification rate (ECAR) were measured (Figure 4A) as indicators of mitochondrial respiration and glycolysis, respectively, in the presence of oligomycin (ATP synthase inhibitor), FCCP (proton ionophore), antimycin A (mitochondrial Complex III inhibitor), and rotenone (mitochondrial Complex I inhibitor) [20,37], as highlighted in Figure 4A, to enable the estimation of basal, ATP-linked, maximal, and non-mitochondrial respiration activities [20]. As previously described in [16], infection with rA2 resulted in a decrease in baseline OCR (Figure 4A, main panel) accompanied by increases in basal ECAR (Figure 4A, inset), indicating inhibition of mitochondrial respiration and a parallel increase in glycolytic metabolism for energy production. Strikingly, infection with rA2:Mmut had little effect on mitochondrial oxidative and glycolytic activity (Figure 4A, inset). Analysis of the pooled data (Figure 4B) revealed significant (*p* < 0.001) decreases (50–60%) in basal, maximal, and ATP-linked OCR in the case of cells infected with rA2, concomitant with a > 60% increase in non-mitochondrial OCR (Figure 4B, black bars), consistent with reduced mitochondrial oxidative phosphorylation (OXPHOS) and increased non-OXPHOS oxygen consumption. In contrast, no such effects were seen for cells infected with rA2:Mmut (Figure 4B, grey bars), consistent with the idea that M is key to inhibiting mitochondrial respiration in RSV infection.

### 3.3. RSV M Is Necessary and Sufficient to Induce Loss of Δψ_m_

We previously showed that RSV infection results in loss of Δψ_m_ [16]. To determine the role of M in this effect, cells were infected with rA2 or rA2:Mmut, and Δψ_m_ was assessed at various times using the Δψ_m_-sensitive dye TMRE [29], together with MitoTracker Deep Red for mitochondrial localisation and the Hoechst dye to define cell nuclei (Figure 5A). The proton ionophore FCCP was used as a control to reveal the effect of maximal dissipation of the Δψ_m_, as indicated by the uniform loss of TMRE fluorescence (Figure 5A, second row of panels). Substantial perinuclear mitochondrial clustering was clearly observed in rA2-infected (Figure 5A, third, fifth, and seventh rows of panels), compared to the rA2:Mmut-infected (fourth, sixth, and last rows of panels) or uninfected (first row of panels) cells. Importantly, rA2 infection resulted in a gradual loss of TMRE fluorescence over 12–24 h p.i. (Figure 5A, third, fifth and seventh rows of panels), whereas rA2:Mmut-infected cells retained TMRE fluorescence throughout the experiment (fourth, sixth, and last rows of panels). Quantification of TMRE fluorescence density (Figure 5B) confirmed the observations, revealing significantly (*p* < 0.01) reduced (over 40%) Δψ_m_ at 18 and 24 h p.i., consistent with the idea that M is necessary to induce a loss of Δψ_m_ in infection.

To confirm the contribution of RSV M to the loss of Δψ_m_, out of the context of viral infection, cells were transfected to express GFP, GFP-M or GFP-Mmut, followed by TMRE staining, and analysis by two-colour flow cytometry (Figure 5C–E). FCCP treatment used as a positive control drastically decreased the number of live cells with high Δψ_m_ (Figure 5C). GFP and TMRE fluorescence were analysed (Figure 5D). A detailed analysis of the GFP-positive cells (Figure 5E) revealed drastically reduced levels of TMRE fluorescence in GFP-M-expressing cells (3009 ± 69.5), compared to GFP-expressing (4250 ± 85.4) and GFP-Mmut-expressing (4243 ± 48.9) cells. The clear implication is that R170T/K172T substitution of M eliminates its ability to impact Δψ_m_ out of the context of viral infection. Together, these results strongly implicate RSV M as being necessary and sufficient to induce loss of Δψ_m_ in infected and transfected cells.

### 3.4. Wild-Type RSV M Contributes to Increased mtROS in Infection

We previously showed that RSV infection results in increased production of mtROS [16]. To confirm M’s contribution to this effect, we stained cells with a reversible sensor of mtROS, flavin-rhodamine redox sensor 2 (FRR2) [33], in parallel with MitoTracker Deep Red to visualise mitochondrial localisation. The oxidised form of FRR2 emits much more strongly at 580 nm upon excitation at 514 nm than at 488 nm, enabling ratiometric live imaging of mtROS production in situ [33]. We used uninfected cells treated with rotenone as a positive control, with strong FRR2 emission able to be visualised in regions colocalising with MitoTracker Deep Red (Figure 6A, third row of panels), indicative of high mtROS. Treatment with mtROS scavenging agent MitoQ [32,38] strongly suppressed the actions of rotenone (Figure 6A, fourth row of panels), confirming that mitochondria were the source of this staining. FRR2 fluorescence increased following virus infection, with clearly evident ratiometric signal (I_(Ex514)_/I_(Ex488)_, far right column) in cells infected with rA2 but not rA2:Mmut (Figure 6A, fifth versus sixth rows). This ratiometric signal was markedly reduced by MitoQ treatment (Figure 6A, bottom two rows of panels), confirming that the observed M-dependent signals were attributable to mtROS. The results from the quantitative analysis of the ratiometric images indicated significantly (*p* < 0.01) increased (over 40%) FRR2 signals by wild-type rA2, but not rA2:Mmut (Figure 6B), consistent with the idea that wild-type M is necessary to induce an enhanced generation of mtROS in infection.

To assess the broader effect of RSV M on intracellular ROS production, we stained rA2- and rA2:Mmut-infected cells with the intracellular ROS indicator 2′,7′-dichlorodihydrofluorescein diacetate (DCF) [34], together with MitoTracker Deep Red for mitochondrial localisation and Hoechst dye for nuclear staining. The treatment of mock-infected cells with rotenone as a positive control resulted in strong DCF staining in regions colocalised with MitoTracker Deep Red (Figure 6C, second row), indicating ROS accumulation in mitochondria. Additional treatment with MitoQ effectively suppressed the actions of rotenone (Figure 6C, third row), confirming the mitochondrial contribution to this ROS staining. Importantly, infection with rA2, but not rA2:Mmut, resulted in a marked increase in the levels of ROS (Figure 6C, compare fourth with fifth row). The increase in ROS levels upon rA2 infection could be reversed by MitoQ post-treatment (Figure 6C, bottom two rows), confirming that the ROS generated in infected cells was largely mitochondrial. Quantitative analysis of DCF fluorescence (Figure 6D) revealed significantly (*p* < 0.001) reduced (over 50%) levels by rA2, but not rA2:Mmut, supporting the conclusion that wild-type M plays a key role in the enhanced production of mtROS and altered oxidative status of host cells evident in RSV infection.

### 3.5. RSV Infection Reduces Host Mitochondrial Gene Expression Dependent on Wild-Type M, and Can Be Reversed by a mtROS Scavenger

Gene profiling has shown that RSV infection impacts the host transcriptome, with M proposed to be a contributor early in infection [7,8,40,41,42,43,44]. To assess M-dependent effects on the expression of host cell mitochondrial genes, RNA from rA2- or rA2:Mmut-infected cells was analysed by RT-qPCR (Figure 7) for a set of genes (see also [8]), in part based on preliminary RNAseq analysis (not shown). Infection with wild-type rA2 at 24 h p.i. did not alter the mRNA levels for centromere protein U (CENPU) and house-keeping genes, including large ribosomal protein (RPLPO) and hypoxanthine phosphoribosyltransferase 1 (HPRT1), but markedly reduced the levels (over 30%) for a number of mitochondrial genes, including those central to mitochondrial metabolism (LIPT2, CYP24A1, PDK4), respiratory Complex I electron transport (NDUFV1, NDUFC2, NDUFA10, NDUFB10), Complex I assembly (ACAD9), and mtROS regulation (PRDX5) (Figure 7A, black bars); this was in stark contrast to non-mitochondrial genes, such as dynein cytoplasmic 1 heavy chain 1 (DYNC1H1) and forkhead box protein O3 (FOXO3), which showed increased (over 50%) mRNA expression in rA2-infected compared to mock-infected cells (Figure 7A, black bars). Infection with rA2:Mmut did not markedly impact the levels of host cell mitochondrial genes (Figure 7A, grey bars), implying that wild-type M is key to the transcriptional downregulation of mitochondrial genes in RSV-infected cells [8].

To test the contribution of mtROS to the above effects, we tested the impact of MitoQ on mitochondrial gene expression (Figure 7B). The addition of the mitochondria-specific ROS scavenger MitoQ for the period 18–24 h p.i. effectively (*p* < 0.05) prevented RSV-induced inhibition of mitochondrial gene expression, as assessed at 24 h p.i. (Figure 7B, compare grey to black bars). Taken together, the results indicate that M’s effects on the host mitochondria in RSV infection extend to the suppression of mitochondrial genes; strikingly, mtROS would appear to be an important contributor to this, since blocking it with MitoQ can reverse this effect. The clear implication of the results here and in our recent published work [8,16] is that M is critical in RSV infection through effects initiated at the level of the nuclear chromatin and nuclear transcription, which in turn strongly impact the host mitochondrial function (see Section 4).

## 4. Discussion

This study shows for the first time that the RSV M protein is sufficient and necessary to remodel host cell mitochondria and perturb their respiratory function, building on previous work demonstrating the effects of RSV infection on mitochondrial morphology/function [16,17], with clear parallel effects of transcriptional inhibition of key mRNAs encoding mitochondrial components [8]. Here, we firstly show that the expression of M, but not other RSV proteins, is sufficient to induce mitochondrial perinuclear clustering, with concomitant loss of Δψ_m_. Mitochondrial remodelling by RSV M is strongly dependent on its central nucleic acid-binding domain, with arginine 170 and lysine 172 being critically important. Mutant RSV carrying R170T/K172T substitutions within M shows a strongly reduced ability to compromise infected host cell mitochondrial respiration and Δψ_m_, and does not induce high levels of mtROS, correlating with the abrogation of chromatin association/transcriptional inhibition [8]. This mutant virus is also impaired in RSV infectious virus production [8], consistent with RSV M’s effects on the host mitochondria being critical to the RSV infectious cycle, presumably through effects at the level of M association with the host chromatin, leading to effects on host genes encoding mitochondrial gene products [8]. The preliminary qPCR analysis of mRNA levels for genes encoding mitochondrial proteins, including those for respiratory Complex I, is consistent with this idea, indicating reduced levels upon infection by wild-type but not the R170T/K172T-substituted virus, implying that part of the mechanism by which mitochondria are impacted in RSV infection is through the M-dependent effects on transcription at the level of the nucleus [8]. The inhibition of mtROS production using MitoQ reduces the effect of RSV infection on mitochondrial gene mRNA levels, implying that mtROS in wild-type infection contributes to the effects on transcription.

Figure 8 summarises the results of this study with a focus on the role of M in remodelling host cell mitochondria, in part through the suppression of the expression of mitochondrial genes (Figure 8). M is known to traffic into the nucleus at the early stages of infection (6–12 h p.i.) through host transporter IMPβ1 [7,9,11,40], with nuclear M key to the inhibition of host transcriptional activity in infected cells [8,10,16,40]. Our results here, together with previous studies [8,17], highlight the fact that mitochondrial genes are among key targets of M action, with the R170T/K172T-substituted M-containing virus showing a reduced impact on host transcription, and thus, the levels of mitochondrial gene products/mitochondrial function. The reduction in host mitochondrial gene mRNA levels clearly correlates closely with the depletion of mitochondrial respiration, loss of Δψ_m_, and mtROS accumulation, implying a close link between the two. The perinuclear clustering of mitochondria would appear to be an initiator/driver of these effects, ultimately contributing to increased nuclear ROS levels [7,8,16]. mtROS feedback to the nucleus may subsequently importantly contribute to modulating the activity of redox-sensitive transcriptional factors, and hence, the levels of redox-sensitive gene mRNAs (Figure 8), potentially amplifying effects later in infection, when M is no longer in the nucleus. M’s role in infection includes impacting the host nuclear-mitochondrial axis; effects at the nuclear chromatin/transcriptional level have strong effects on the host mitochondria, and in turn, induced ROS production by the mitochondria is likely to impact nuclear transcription through nuclear transcription factors, such as nuclear factor erythroid 2 (NFE2)-related factor 2 (Nrf2) [7,8,45,46].

Importantly, this study delineates the role of M in RSV’s specific effects on host cell mitochondria for the first time. It is clear that the ectopic expression of M alone can induce mitochondrial perinuclear clustering, with wild-type M critical to the same effects in the context of RSV infection. Previously, we showed that the mitochondrial perinuclear clustering in RSV-infected cells is dynein- and microtubule-dependent [16]. Here, we show that RSV infection, dependent on M, results in a significant increase in mRNA levels for cytoplasmic dynein (DYNC1H1), underlining the fact that M’s effects on nuclear gene transcription extend to genes other than those encoding mitochondrial gene products; increased dynein expression may potentially contribute to the movement of mitochondria to the MTOC in RSV-infected cells [16].

In conclusion, this study establishes for the first time that RSV M, dependent on arginine/lysine residues 170/172 in its central nucleic acid-binding domain, is a key factor in the remodelling of host cell mitochondria, with impacts on host mitochondrial function. Significantly, M-dependent effects on mitochondria are critical to RSV virus production, underlining their importance with respect to the RSV infectious cycle. The fact that the mtROS scavenging agent MitoQ can limit RSV M’s effects on mitochondria further highlights the M-mitochondrial interface as an exciting target for the development of anti-RSV strategies. In this context, MitoQ is available as an antioxidant “supplement” deemed safe for human use, and has previously been shown to have efficacy in limiting RSV infection in a primary human bronchial epithelia cell model of RSV infection, as well as a mouse model, where it reduced both viremia and systemic inflammation [16]. The FDA-approved therapeutic monoclonal antibodies Palivizumab and the relatively new Nirsevimab [47,48] are currently in vogue to prevent RSV infection in newborns, but their applicability to treat/protect the elderly remains unclear. Along with Nirsevimab and Verdinexor [49], an inhibitor of Crm1 analogous to leptomycin B, which inhibits RSV infectious virus production [11], MitoQ may indeed be an interesting prospect as an emerging anti-RSV treatment agent. Finally, even though conventional vaccine development has been notoriously difficult for RSV, the recombinant RSV Mmutant virus characterised here for lack of pathological effects on the host mitochondria (and previously by Li et al. [8] for impaired transcriptional inhibition), may be considered as having potential as an attenuated virus vaccine candidate; as already mentioned, the immune responses to the virus are clearly altered in mice (e.g., increased RANTES production), in parallel with reduced viremia/lung inflammation [8]. Further investigation of antiviral/vaccine strategies based on the work here and elsewhere [8,16] are the focus of future work in this laboratory.

## Figures and Tables

**Figure 1 cells-12-01311-f001:**
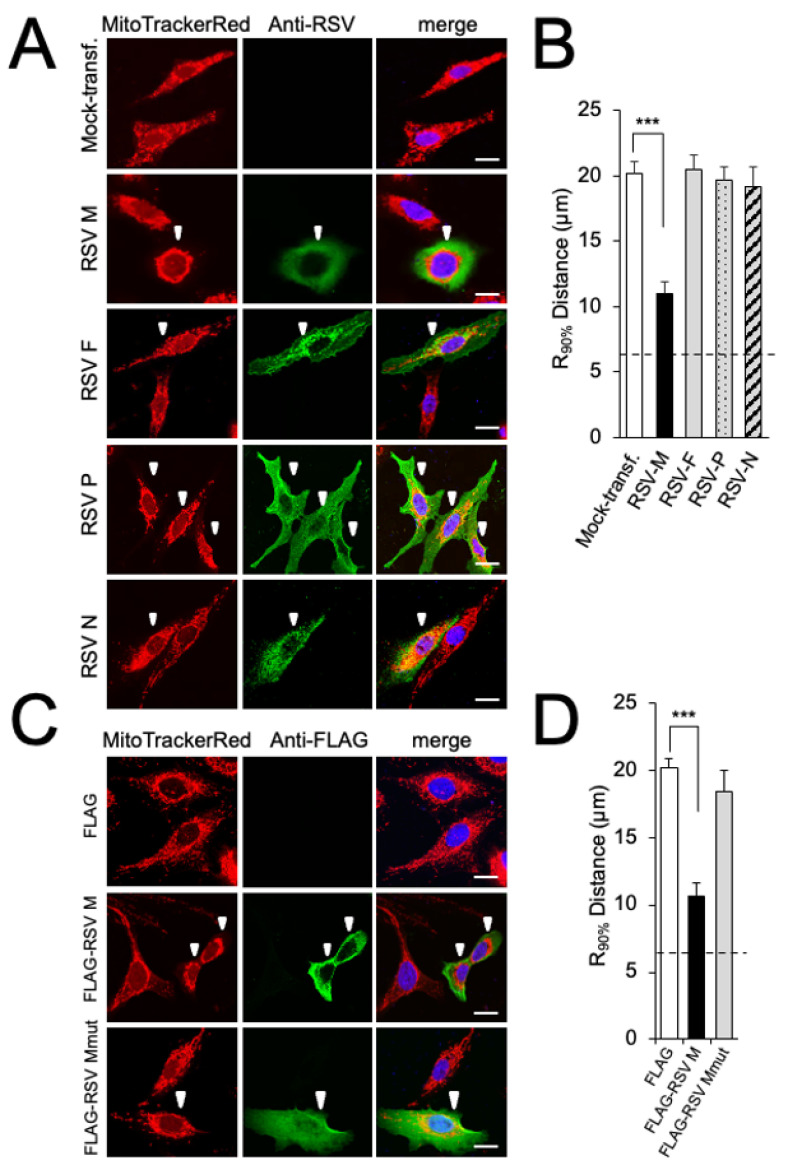
RSV matrix protein (M) is sufficient to induce mitochondrial perinuclear clustering. A549 cells were transfected to express the indicated constructs or mock-transfected, then treated with MitoTrackerRed 24 h later, and fixed and stained using anti-RSV (**A**,**B**) or -FLAG (**C**,**D**) antibodies (green) and DAPI (blue for nuclei). (**A**,**C**) Cells were imaged by CLSM. Merge panels overlay all three stains. White arrowheads highlight RSV-transfected cells. Scale bar = 10 µm. (**B**,**D**) The perinuclear radial distribution for MitoTrackerRed staining (R_90%_; see Section 2) was calculated from images such as those in (**A**) or (**C**). Results represent the mean ± SEM for *n* = 3 independent experiments, each of which analysed 25–30 cells per sample; *** *p* < 0.001. The dashed line represents the average nuclear radius.

**Figure 2 cells-12-01311-f002:**
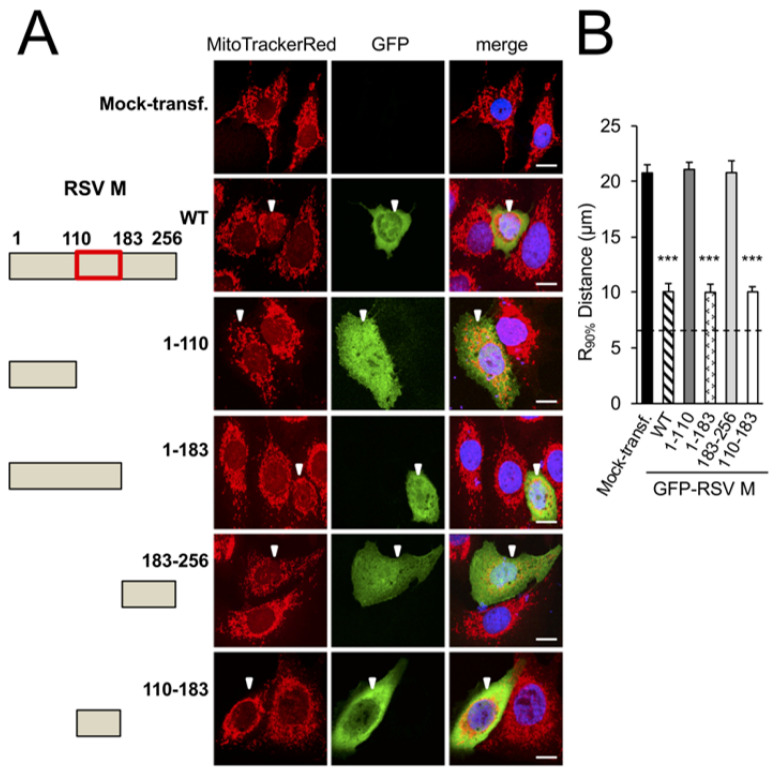
Arginine 170 and lysine 172 within the central nucleic acid-binding domain of M are important for mitochondrial perinuclear clustering. A549 cells were mock-transfected or transfected to express the in653dicated GFP-RSV M fusion construct, then treated with MitoTrackerRed 24 h later, and fixed and stained using anti-GFP antibody (green) and DAPI (blue for nuclei). (**A**) Schematic representations of RSV M in the fusion constructs used (left). Numbers refer to the amino acid sequence of M, with the central nucleic acid-binding domain shown as a red box. Cells were imaged by CLSM as per Figure 1A (right). White arrowheads denote infected cells, scale bar = 10 µm. (**B**) R_90%_ analysis was as per Figure 1B. (**C**) Schematic representation (left) showing mutational subsitutions within the central nucleic acid-binding domain of RSV M (residues 110–181) in the fusion constructs, as per (**A**), with substitutions of basic residues highlighted in red. (**D**) R_90%_ analysis was as per Figure 1B. *** *p* < 0.001, ** *p* < 0.01.

**Figure 3 cells-12-01311-f003:**
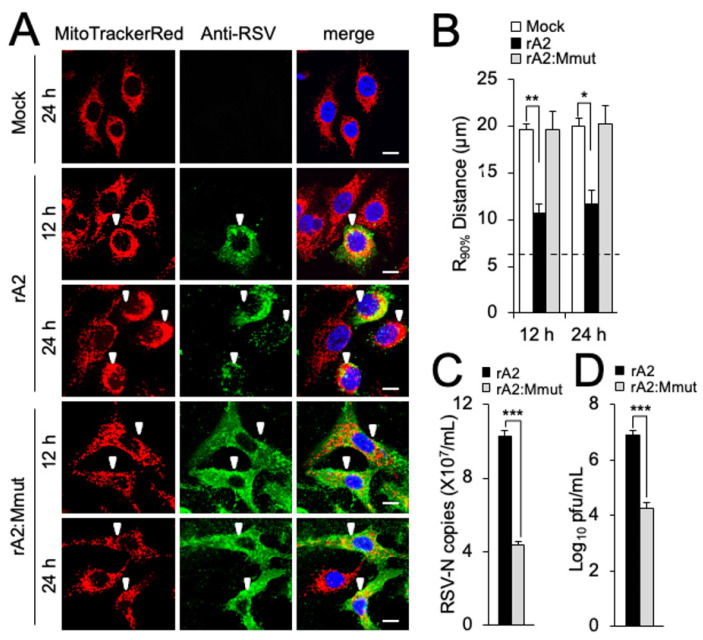
Mitochondrial perinuclear clustering in infected cells is critical to RSV virus production. A549 cells were mock-, rA2- or rA2 M:R170T/K172T (rA2:Mmut)-infected (MOI 3) for the times indicated and then subjected to (**A**,**B**) fixation/immunostaining and R_90%_ analysis as per Figure 1A,B, where white arrowheads denote infected cells (scale bar = 10 µm); or (**C**,**D**) quantification of cell-associated (**C**) virus genomes by qPCR, and (**D**) infectious virus by plaque assay. Results shown represent the mean ± SEM from 3 independent experiments assayed in triplicate. *** *p* < 0.001, ** *p* < 0.01, * *p* < 0.05.

**Figure 4 cells-12-01311-f004:**
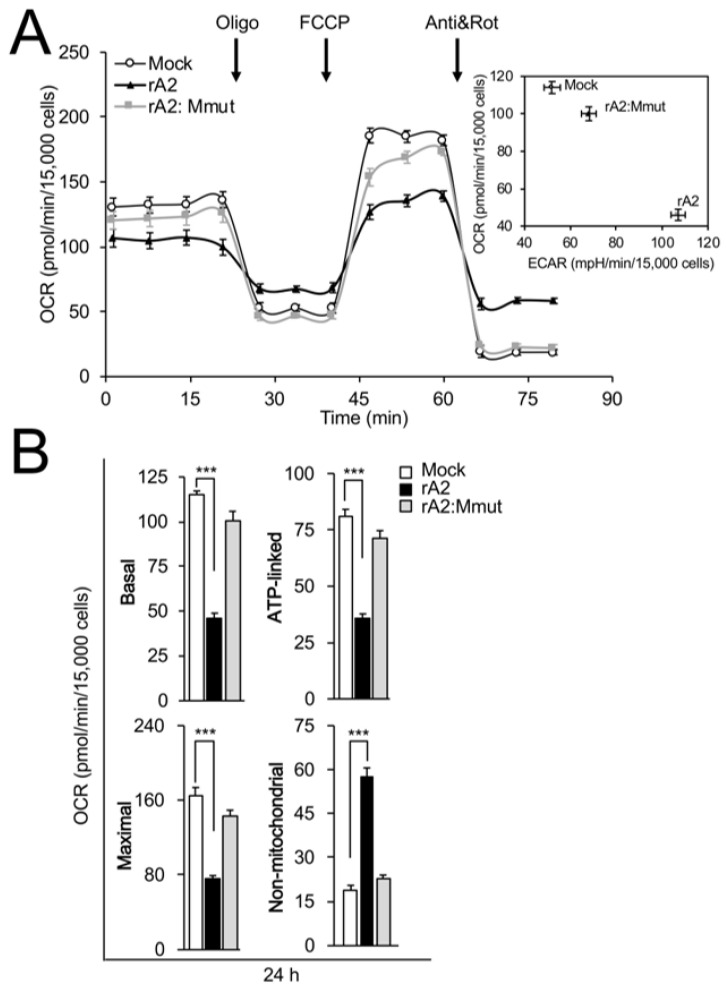
RSV M is necessary for inhibition of mitochondrial respiration in infected cells. The bioenergetic status of A549 cells infected without (mock) or with rA2 or rA2:Mmut (MOI 3) was assessed 24 h p.i. using the Seahorse XF96 Extracellular Flux Analyser. (**A**) Example of a typical oxygen consumption rate (OCR) obtained in these experiments. OCR was measured in real time upon sequential additions of ATP synthase inhibitor oligomycin (Oligo, 1 µM), proton ionophore FCCP (1 µM), mitochondrial Complex III inhibitor antimycin A (Anti, 1 µM), and mitochondrial Complex I inhibitor rotenone (Rot, 1 µM). Inset: Plot of basal OCR (a measure of mitochondrial respiration) and extracellular acidification rate (ECAR, an indicator of glycolysis) in infected A549 cells. (**B**) Pooled data for mitochondrial respiratory parameters of basal, ATP-linked, maximal and non-mitochondrial respiration (see Section 2). Results represent the mean ± SEM for *n* = 3 independent experiments, each performed in triplicate. *** *p* < 0.001 compared to the mock-infected cells.

**Figure 5 cells-12-01311-f005:**
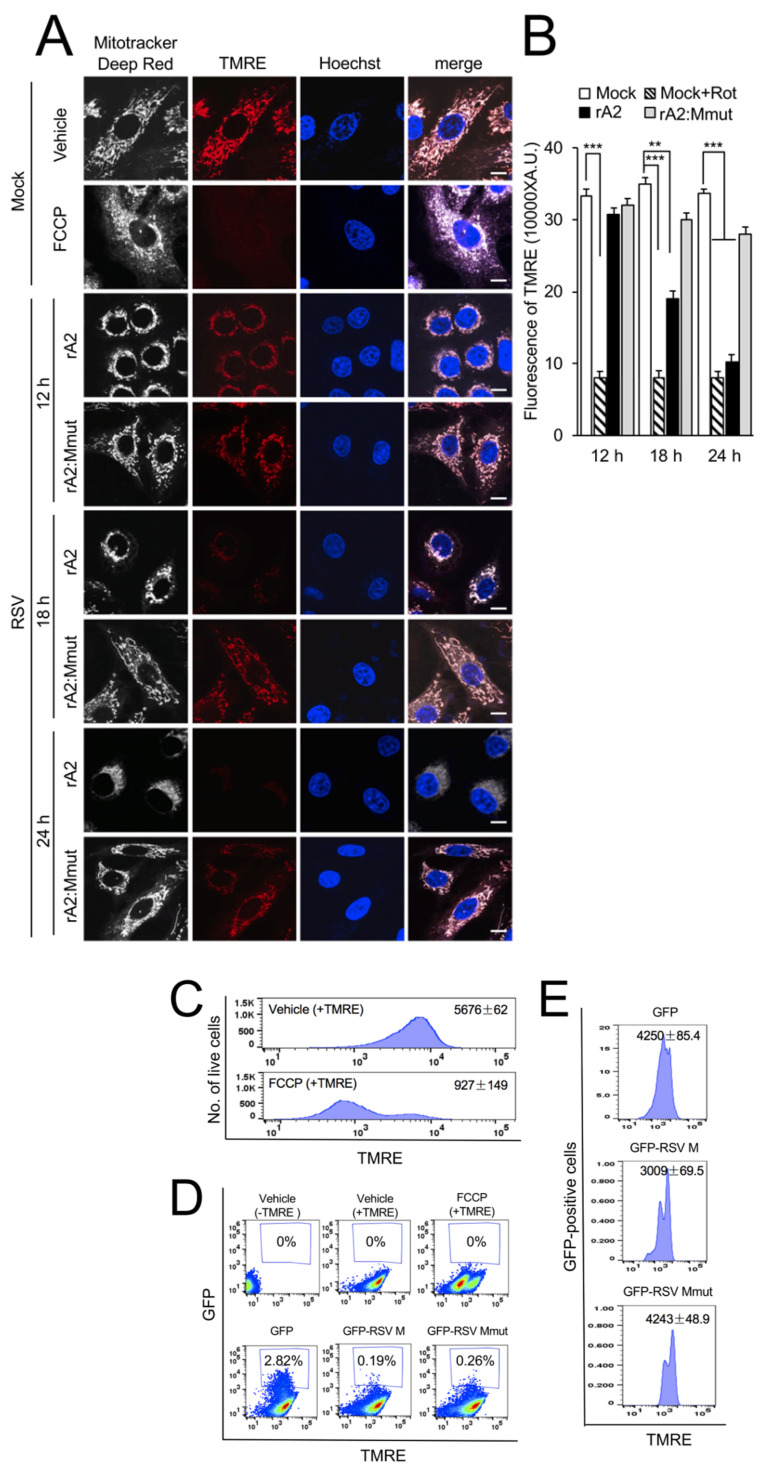
RSV M is necessary and sufficient to induce loss of Δψ_m_ in RSV infected/transfected cells. (**A**,**B**) A549 cells were infected without (mock) or with rA2 or rA2:Mmut (MOI 3) for 12–24 h, as indicated, followed by treatment with DMSO as vehicle or FCCP (5 µM, 10 min). Cells were stained for the Δψm-sensitive dye TMRE (red; 50 nM) and MitoTracker Deep Red (white; 100 nM) for the final 15 min and Hoechst nucleic acid dye (blue; 5 µg/mL) for 5 min prior to live imaging by CLSM. (**A**) Merge panels overlay all three stains. Scale bar = 5 µm. (**B**) Quantitative analysis of the average fluorescence density of the TMRE staining was performed using Fiji software. Results represent the mean ± SEM (*n* = 3 independent experiments), as per (**A**). *** *p* < 0.001, ** *p* < 0.01 compare to the mock vehicle-treated control. (**C**–**E**) FACS analysis of single-cell suspensions stained with TMRE. (**C**) Cells were mock-transfected followed by treatment with DMSO as vehicle or FCCP (5 µM, 10 min), or (**D**,**E**) were transfected with GFP, GFP-RSV M or GFP-RSV M:R170T/K172T (GFP RSV-Mmut) for 24 h before TMRE staining as per (**A**). (**D**) The percentages of GFP-positive cells (in blue boxes, 50,000/sample) were determined by GFP (Ex/Em: 488/540 nm) compared to TMRE (Ex/Em: 561/580 nm) fluorescence using FlowJo. (**E**) TMRE fluorescence analysis of GFP-positive cells. Results represent the mean ± SEM for *n* = 3 independent experiments, each of which analysed 15–20 cells per sample.

**Figure 6 cells-12-01311-f006:**
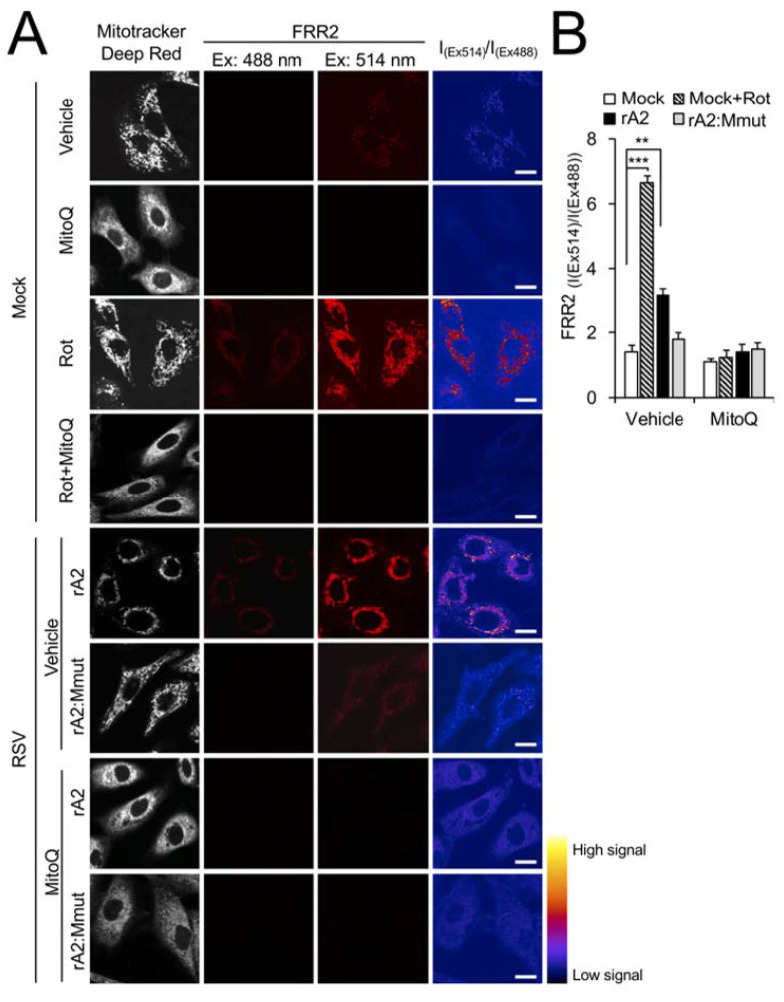
Wild-type RSV M contributes to generation of mtROS in RSV infection. A549 cells were infected without (mock) or with (rA2 or rA2:Mmut (MOI 3) for 12 h prior to the treatment, as indicated: the mitochondrial Complex I inhibitor rotenone (Rot, 0.5 µM), the mitochondria-specific ROS scavenger mitoquinone mesylate (MitoQ, 1 µM), NCZ (17 µM) or DMSO as a vehicle for 2 h, or Rot (0.5 µM, 2 h) with the addition of MitoQ (1 µM) for an additional 2 h (Rot + MitoQ). (**A**) Cells were stained for MitoTracker Deep Red (white; 100 nM, 15 min) and the mitochondria-specific ROS probe flavin-rhodamine redox sensor 2 (FRR2, red; 2 µM, 15 min prior to imaging). Colocalization for FRR2 staining at either Ex488 or Ex514 and MitoTracker Deep Red was >90% (Pearson correlation coefficient [39]) across all samples (25–30 cells/sample). The ratiometric output images of I_(Ex514)_/I_(Ex488)_ (far right) were calculated by pixel-wise division of FRR2 emission (580 ± 20 nm) images acquired using excitation at 514 nm (third column) or 488 nm (second column), and are represented in pseudo-colour (intensity colour bar displayed lower right). Live cell imaging was performed by resonant scanning CLSM. Results are typical of 3 independent experiments. Scale bar = 10 µm. (**B**) FRR2 (I_(Ex514)_/I_(Ex488)_) was calculated for the mitochondrial regions defined by MitoTracker Deep Red staining in the I_(Ex514)_/I_(Ex488)_ images, such as those in **A**, using a custom CellProfiler pipeline (see Section 2). Results represent the mean ± SEM for *n* = 3 independent experiments, where each experiment analysed 25–30 cells per sample, *** *p* < 0.001, ** *p* < 0.01 compare to the mock vehicle-treated control. (**C**,**D**) Cells were stained with MitoTracker Deep Red as for **A**, in addition to Hoechst nucleic acid dye (blue; 5 µg/mL) and the cellular ROS indicator 2′,7′-dichlorodihydrofluorescein diacetate (DCF, magenta; 2.5 µM) for 5 min prior to live cell imaging by CLSM. (**C**) Merge panels overlay all three stains. In all panels, scale bar = 10 µm. (**D**) Quantitative analysis of average fluorescence density of DCF by Fiji software. Results represent the mean ± SEM for 3 independent experiments as per (**C**). *** *p* < 0.001 compare to the mock vehicle-treated control.

**Figure 7 cells-12-01311-f007:**
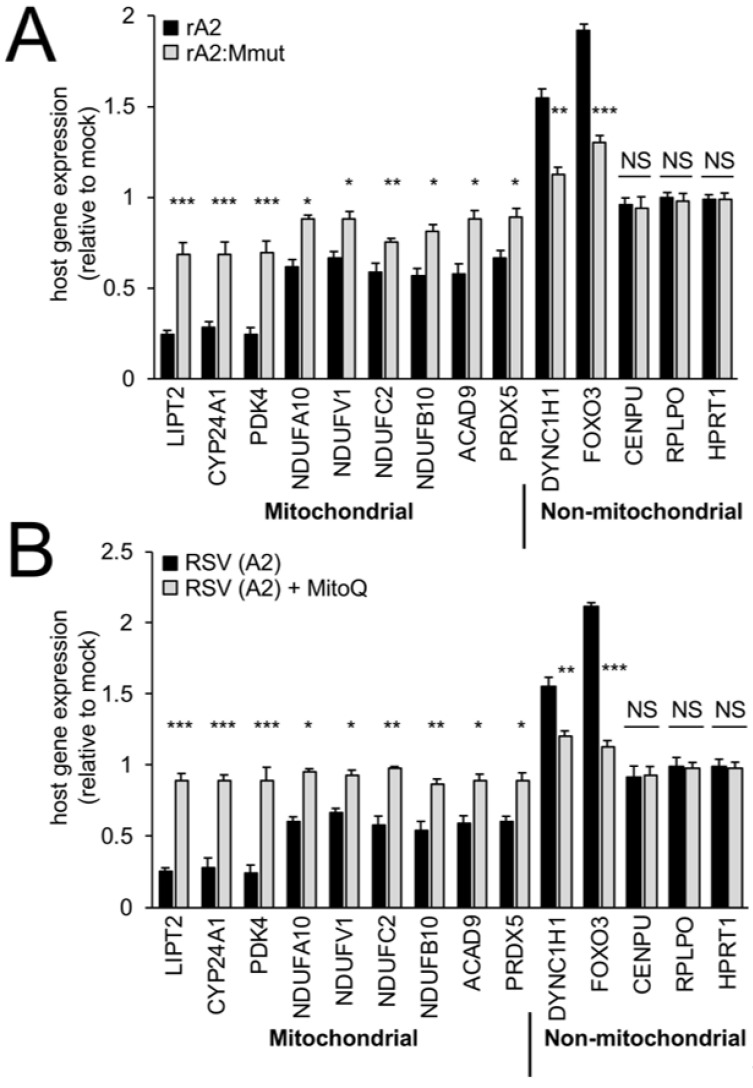
RSV infection reduces host mitochondrial gene expression dependent on wild-type M, and can be reversed by a mtROS scavenger. A549 cells were infected without (mock) or with (**A**) rA2- or rA2:Mmut-infected (MOI 3) for 24 h; or (**B**) RSV A2-infected (MOI 3) followed by treatment with DMSO (vehicle) or MitoQ (1 μM) over 18–24 h, prior to cell lysis, RNA preparation, and RT-qPCR analysis, as per the Section 2. All host genes (see Section 2) were standardised against endogenous control genes, RPLPO and HPRT1, and were expressed relative to the corresponding genes in the mock-infected cells. Results represent the mean ± SEM (*n* = 3 independent experiments, each performed in triplicate). *** *p* < 0.001, ** *p* < 0.01, * *p* < 0.05 compared to rA2- or RSV A2-infected cells.

**Figure 8 cells-12-01311-f008:**
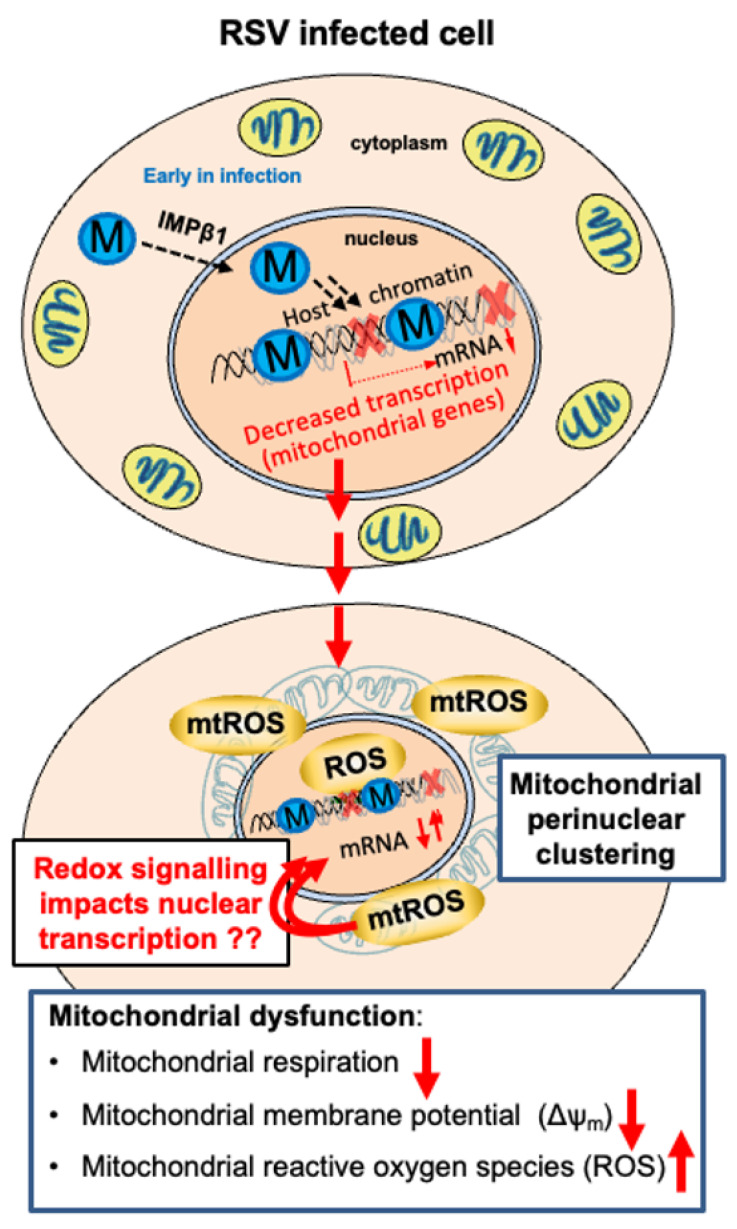
Model for the role of M in remodelling host mitochondria and the nuclear-mitochondrial axis in RSV infection. RSV M traffics into the nucleus of infected cells early in infection to access the chromatin [8] and inhibit the expression of nucleus-encoded mitochondrial genes. This contributes to the perinuclear clustering of the mitochondria, with accompanying impaired mitochondrial respiration, loss of mitochondrial membrane potential, and enhanced production of mitochondrial reactive oxygen species (mtROS). The latter likely adds a further level of amplification through nuclear ROS (see [16]) impacting redox-sensitive transcription factors (see [7]), with nuclear transcription again impacted [8]. Later in infection (not shown), M exits the nucleus and enters the cytoplasm dependent on Crm1 [11], coinciding with a reversal of the functional effects on mitochondria (see [16]) to participate in virus assembly.

## Data Availability

Data supporting reported results can be obtained upon a request to the corresponding author (D.A.J.).

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
