# Peer review of "Respiratory Syncytial Virus Matrix Protein Is Sufficient and Necessary to Remodel Host Mitochondria in Infection"

_cells, 2023, doi:10.3390/cells12091311_

Round 1

Reviewer 1 Report

This is a highly interesting paper with new data on the role of RSV M protein in RSV infection.

The only matter of substance that is misleading and should be corrected in the manuscript is the claim that "there is no efficacious antiviral strategy to combat the virus". There are currently two monoclonal antibodies, Palivizumab and Nirsevimab to prevent RSV infection in the newborn, especially if they are prone to severe outcomes. The recently approved Nirsevimab recognising the pre-fusion form of the F protein maybe a game changer what comes to applicability. 

On the other hand, authors could elaborate a little more on the possibility to use RSV with mutated M alone or with other attenuations as a live vaccine or a drug restoring the mitochondrial function during RSV infection. 

As a matter of readability, please spell out the acronyms, when they first appear in the main text.

In Figures, please indicate "anti-RSV" instead of "RSV", when antibodies were used to stain the viral protein(s).

Author Response

We thank the Reviewer for the positive comments. We have addressed the minor concerns as follows (our changes in italics).

The only matter of substance that is misleading and should be corrected in the manuscript is the claim that "there is no efficacious antiviral strategy to combat the virus". There are currently two monoclonal antibodies, Palivizumab and Nirsevimab to prevent RSV infection in the newborn, especially if they are prone to severe outcomes. The recently approved Nirsevimab recognising the pre-fusion form of the F protein maybe a game changer what comes to applicability. 

We have modified lines 34-35 (Introduction), as appropriate, to reflect the current situation. Nirsavimab is certainly an exciting prospect as prophylactic for newborns, but the extent to which this treatment will stand the test of time in clinical use with infants, and whether it has applicability in other contexts (eg. in the elderly) is not yet clear. To satisfy the Reviewer, we include discussion of Nirsevimab in the Discussion section (lines 776-782). We thank the Reviewer.

On the other hand, authors could elaborate a little more on the possibility to use RSV with mutated M alone or with other attenuations as a live vaccine or a drug restoring the mitochondrial function during RSV infection. 

To satisfy the Reviewer, we expand on these possibilities in the final Discussion (Lines 782-788). We thank the Reviewer.

As a matter of readability, please spell out the acronyms, when they first appear in the main text.

We have made sure all acronyms are spelled out where they first appear - we thank the Reviewer.

In Figures, please indicate "anti-RSV" instead of "RSV", when antibodies were used to stain the viral protein(s).

We have modified Figures 1 and 3 as required (we also changed “FLAG” to anti-FLAG) as requested. We thank the Reviewer.

We thank the Reviewer for the important suggestions, which have enhanced our manuscript.

Reviewer 2 Report

A brief summary:

The aim of the paper was to establish the exclusive role of the RSV matrix protein (M) in the redistribution of mitochondria with compromised respiratory activities and increased reactive oxygen species (ROS) generation during RSV infection. The results obtained highlight unique ability of M to remodel host cell mitochondria and show its critical role in RSV infection. The authors believe that this work represent a novel, potential target for future anti-RSV strategies, which is of significant importance due to the lack of effective antiviral drugs for the treatments for RSV infection..
      The presented manuscript is clear, relevant mostly for Virology field and presented in a well-structured manner. The cited references match the content of the article; include four (10.6%) recent publications (within the last 5 years) and eight self-citations of 47 references.

     The manuscript scientifically sound and the experimental design appropriate to test the hypothesis. The results are reproducible based on the details given in the methods section.

The figures/tables/images/schemes are appropriate, they properly show the data. They are easy to interpret and understand. The data as a rule interpreted appropriately and consistently throughout the manuscript. Methods of statistical processing of results, where applicable, are justified, with a value of p < 0.05 considered to be statistically significantly different.

The conclusions are consistent with the evidence presented.

       The work is performed at a modern, very high methodological level and beautifully illustrated. Using a combination of redox/membrane potential -sensitive dyes, high-resolution imaging/flow cytometric analysis, reverse transcription quantitative PCR and bioenergetics analyses, it proved possible to establish that ectopically expressed M, but not other RSV proteins, is able to induce mitochondrial perinuclear clustering. The results interpreted appropriately and conclusions justified and supported by the results.

      I have some questions, which I would like to receive as an answer in the Discussion. The authors indicate, that the “rA2:Mmut virus showing reduced viral replication and production underlines the importance of M-dependent mitochondrial redistribution to RSV infection” (pp.332-334). The question arises, whether the M-dependent mitochondrial redistribution is a dose-dependent effect and at a low content of the replicating virus, as is observed with the rA2:Mmut virus, mitochondrial redistribution might not occur? Have the authors used other, less reproductive strains of RSV in the work to assess the degree of redistribution of mitochondria during such infection? In the absence of such data, I would change the title of the work somewhat, removing the words “sufficient and necessary” for “essentially important” or otherwise.

I think also, that it would be useful to develop in more detail the idea expressed by the authors, that “MitoQ can limit RSV M’s effects on mitochondria further highlights the M-mitochondrial interface as an exciting target for the development of anti-RSV strategies in the future” (pp. 625-627) to identify the hypotheses more carefully.

I would advise the authors also to remove links to their own figures from the Discussion, so that the impression of repeating the results not created. It would be useful to supplement this section of the manuscript with the latest data on the development of new anti-RSV drugs, such as Verdinexor.

Author Response

We thank the Reviewer for the very positive comments, and constructive suggestions to enhance the manuscript, which we have taken on board as far as possible (see responses inbox italics below).

I have some questions, which I would like to receive as an answer in the Discussion. The authors indicate, that the “rA2:Mmut virus showing reduced viral replication and production underlines the importance of M-dependent mitochondrial redistribution to RSV infection” (pp.332-334). The question arises, whether the M-dependent mitochondrial redistribution is a dose-dependent effect and at a low content of the replicating virus, as is observed with the rA2:Mmut virus, mitochondrial redistribution might not occur? Have the authors used other, less reproductive strains of RSV in the work to assess the degree of redistribution of mitochondria during such infection? In the absence of such data, I would change the title of the work somewhat, removing the words “sufficient and necessary” for “essentially important”or otherwise.

We thank the Reviewer, but point out that lines 332-334 were actually duplicated text which have now been deleted. We believe “sufficient and necessary” is completely appropriate as the title of our submission. “Sufficient” in this context means that M is “enough” - clearly, the transfection experiments in the present submission (Fig. 1) clearly support this conclusion; further, cells transfected to express mutated M (mutations that impair transcriptional inhibition – Ref. 8) are impaired in this response. “Necessary” means “required” – clearly, if another RSV protein is transfected, no effect on mitochondria is seen; similarly, if M is mutated in the RSV infection or transfection context, no effect on mitochondria is seen. These results are “all or nothing” – quite simply, M and M alone is responsible, hence the title of the manuscript.

(In terms of the interest of the Reviewer in the question of the “concentration” or “threshold amount” of M required for the effect, the Reviewer may not be aware of previousy published data. Ref. 8 demonstrates dose-dependent effects of nuclear levels of RSV M on transcriptional inhibition in both RSV infected and GFP-M-transfected cells (Figure 1), as well as in vitro (Supp Figure S1) – RSV Mmut infected and GFP-Mmut-transfected cells lack activity (clearly, Wt M is sufficient and necessary). Importantly, these data parallel perfectly the dose-dependent effects on mitochondrial respiration in Ref. 16 (Supp Figure 1), underlining the link between transcriptional inhibition and effects on mitochondria. We thank the Reviewer for the interest.)

I think also, that it would be useful to develop in more detail the idea expressed by the authors, that “MitoQ can limit RSV M’s effects on mitochondria further highlights the M-mitochondrial interface as an exciting target for the development of anti-RSV strategies in the future” (pp. 625-627) to identify the hypotheses more carefully.

We thank the Reviewer – we have expanded the comments on MitoQ and anti-RSV strategies in the revised Discussion (lines 773-782).

I would advise the authors also to remove links to their own figures from the Discussion, so that the impression of repeating the results not created. It would be useful to supplement this section of the manuscript with the latest data on the development of new anti-RSV drugs, such as Verdinexor.

We thank the Reviewer – we have included verdinexor etc. in the Discussion (Lines 779-781).

We thank the Reviewer again for contributing significantly to improving our submission.

Reviewer 3 Report

This group previously showed that RSV remodeled host cell mitochondria and affected respiratory function and ROS production. In this paper, the authors show that the impacts on mitchondria are dependent upon the matrix protein, in particular matrix residues 170/172. A comprehensive series of assays demonstrates that M alone is sufficient to induce mitochondrial clustering, and that impairment of mitochondrial respiration, loss of mitochondrial membrane potential, mtROS generation, and reduction in mitochondrial gene expression, are all dependent on the matrix protein. It is also shown that the mitochondrial clustering plays an important in virus production, reducing viral yield by 2-3 logs when matrix residues 170/172 are mutated. The final figure presents a model in which the suppression of mitochondrial gene expression is suggested to underlie all of the noted effects. Together the work provides a compelling story of the role of matrix protein in mitochondrial remodeling and function. Although the mechanisms remain to be discovered, the study points to a critical role for matrix protein and mitochondrial remodeling in the RSV life cycle.

Major comments

*   Fig 1 C and D are not explained in the text

*   Lines 328-334 appear out of place

*  Since R170T/K172T resulted in a 2-3 log reduction in viral titer, how did the authors generate suffcient M mutant virus for moi-3 infections?

*  Fig 5: it is unclear how the cell numbers in Fig 5E are derived. What do the bottom panels of Fig 5D represent? Is the Y-axis level of GFP? If so, and GFP-RSV Mmut has more GFP-positive cells in Fig. 5E, then GFP-RSV-M and GFP-RSV-Mmut cannot have the similar values in Fig 5D (0.19% and 0.26%).

Minor comments

* Line 406: please explain here how TMRE works.

Author Response

We thank the Reviewer for the rigorous review/probing comments. Our responses/changes in the text are detailed in italics below.

Major comments

*   Fig 1 C and D are not explained in the text

We have amended this error – see new lines 288-300. We thank the Reviewer for pointing out this omission.

*   Lines 328-334 appear out of place

We thank the Reviewer – somehow repeated text from the Results appeared in Fig2 legend – this has been deleted.

*  Since R170T/K172T resulted in a 2-3 log reduction in viral titer, how did the authors generate suffcient M mutant virus for moi-3 infections?

The details of virus production have been published (Ref 8 - Li et al 2021) – lines 88-90 of the Methods section have been expanded to satisfy the Reviewer, and now include this citation. In brief, standard approaches are used (large numbers of cells etc.) with all viral stocks are titrated for PFU before use so that identical MOIs are used in all experiments.

*  Fig 5: it is unclear how the cell numbers in Fig 5E are derived. What do the bottom panels of Fig 5D represent? Is the Y-axis level of GFP? If so, and GFP-RSV Mmut has more GFP-positive cells in Fig. 5E, then GFP-RSV-M and GFP-RSV-Mmut cannot have the similar values in Fig 5D (0.19% and 0.26%).

Perhaps the Reviewer has misunderstood - there is no disparity in the data in Fig. 5D and 5E. The Reviewer is correct that there is not a significant difference in transfection numbers/percentages – the wild type M-GFP construct yields around 27% less transfectants than the mutant construct (in both 5D and 5E – D shows percentages, and E shows the cell numbers from D). The slightly reduced numbers for wt are likely attributable to wt M being able to suppress nuclear transcription robustly, whereas the mutant is not (see Ref 8).

Minor comments

* Line 406: please explain here how TMRE works.

TMRE is explained in some detail in the Methods section (2.6) and again in the Results section (3.4), in conjunction with the FCCP Control. Appropriate references are included.

We thank the Reviewer for contributing importantly to our submission.